# Optimal detection of the feature matching map in presence of noise and outliers

## Abstract

We consider the problem of finding the matching map between two sets of $d$-dimensional vectors from noisy observations, where the second set contains outliers. The matching map is then an injection, which can be consistently estimated only if the vectors of the second set are well separated. The main result shows that, in the high-dimensional setting, a detection region of unknown injection may be characterized by the sets of vectors for which the inlier-inlier distance is of order at least $d^{1/4}$ and the inlier-outlier distance is of order at least $d^{1/2}$. These rates are achieved using the estimated matching minimizing the sum of logarithms of distances between matched pairs of points. We also prove lower bounds establishing optimality of these rates. Finally, we report the results of numerical experiments on both synthetic and real world data that illustrate our theoretical results and provide further insight into the properties of the estimators studied in this work.

## 1 Introduction

Finding the best match between two clouds of points is a problem encountered in many real problems. In computer vision, one can look for correspondences between two sets of local descriptors extracted from two images. In text analysis, one can be interested in matching vector representations of the words of two similar texts, potentially in two different languages. The goal of the present work is to gain theoretical understanding of the statistical limits of the matching problem.

In the sequel, we use the notation $[n] = \{1, \ldots, n\}$ for any integer $n$, and define $\|\cdot\|$ as the Euclidean norm in $\mathbb{R}^d$. Assume that two independent sequences $\boldsymbol{X} = (X_i; i \in [n])$ and $\boldsymbol{Y} = (Y_i; i \in [n])$ of independent vectors are generated such that $X_i$ and $Y_i$ are drawn from the same distribution $P_i$ on $\mathbb{R}^d$, for every $i \in [n]$. The statistician observes the sequence $\boldsymbol{X}$ and a shuffled version $\boldsymbol{X}^\#$ of the sequence $\boldsymbol{Y}$. More precisely, $\boldsymbol{X}^\#$ is such that $X_i^\# = Y_{\pi^*(i)}$ for some unobserved permutation $\pi^*$. The goal of matching is to estimate the permutation $\pi^*$ from data $(\boldsymbol{X}, \boldsymbol{X}^\#)$. In the case of Gaussian distributions $P_i$, this problem has been studied in (Collier and Dalalyan, 2013, 2016). Clearly, consistent estimation of the matching map $\pi^*$ is impossible if there are two data generating distributions $P_i$ and $P_j$ that are very close. In (Collier and Dalalyan, 2013, 2016), a precise quantification of the separation between these distributions is given that enables consistent estimation of $\pi^*$. Furthermore, it is shown that the permutation estimator minimizing the sum of logarithms of pairwise distances between the elements of $\boldsymbol{X}$ and the elements of the shuffled version $\boldsymbol{X}^\#$ is an optimal estimator of $\pi^*$.

In this paper, we extend the model studied in (Collier and Dalalyan, 2016) to the case when the set $\boldsymbol{X}^\#$ is contaminated by outliers. The number of outliers is supposed to be known and is equal to $m - n$, where $n = |\boldsymbol{X}|$ and $m = |\boldsymbol{X}^\#|$ are the sizes of considered two sequences, however the indices of the

|  | Without outliers (Collier and Dalalyan, 2016) | With outliers this paper |
|---|---|---|
| known $\boldsymbol{\sigma}^{\#}$ or all equal $\boldsymbol{\sigma}^{\#}$s | LSNS is optimal $\bar{\kappa} \gtrsim (d \log n)^{1/4}$ | LSNS is optimal [Thm. 1] $\bar{\kappa}_{\text{in-in}} \wedge \bar{\kappa}_{\text{in-out}} \gtrsim (d \log(nm))^{1/4}$ |
| unknown $\boldsymbol{\sigma}^{\#}$ $\sigma_{\max}/\sigma_{\min} \leq C$ | LSL is optimal $\bar{\kappa} \gtrsim (d \log n)^{1/4}$ | LSL is optimal [Thm. 4] $\bar{\kappa}_{\text{in-in}} \gtrsim (d \log(nm))^{1/4}$ & $\bar{\kappa}_{\text{in-out}} \gtrsim d^{1/2}$ |
| unknown $\boldsymbol{\sigma}^{\#}$ arbitrary | | LSL is optimal [Thm. 2, 3] $\bar{\kappa}_{\text{in-in}} \wedge \bar{\kappa}_{\text{in-out}} \gtrsim d^{1/2}$ |

Table 1: A brief overview of the contributions in the high-dimensional regime $d \geq c \log n$. The table provides the condition on the normalized inlier-inlier distance $\bar{\kappa}_{\text{in-in}}$ and inlier-outlier distance $\bar{\kappa}_{\text{in-out}}$, making it possible to consistently detect the matching map between two sets of $d$-dimensional vectors. LSL and LSNS refer to least sum of logarithms and least sum of normalized squares, respectively.

outliers are unknown. All the distributions are assumed in this paper to be spherical Gaussian, although all the probabilistic tools used in the proofs have their sub-Gaussian counterparts. Thus, we consider that two spherical Gaussian distributions [1] $P_1 = \mathcal{N}_d(\mu_1, \sigma_1^2 \mathbf{I}_d)$ and $P_2 = \mathcal{N}_d(\mu_2, \sigma_2^2 \mathbf{I}_d)$ are well separated if the "distance to noise ratio" $\kappa(P_1, P_2) = \|\mu_1 - \mu_2\|/\sqrt{\sigma_1^2 + \sigma_2^2}$ is large. Main findings of (Collier and Dalalyan, 2016), in terms of smallest separation distance $\bar{\kappa} = \min_{i \neq j} \kappa(P_i, P_j)$ are summarized in the second columns of Table 1. Likewise, the last column of the table provides a summary of the contributions of the present paper in terms of $\bar{\kappa}_{\text{in-in}} = \min_{i \neq j} \kappa(P_i, P_j)$ and $\bar{\kappa}_{\text{in-out}} = \min_{i,j} \kappa(P_i, Q_j)$, where $Q_1, \ldots, Q_{m-n}$ are the distributions of the outliers.

An unexpected finding of this work is that the "degree" of heteroscedasticity of the model has a strong impact on the separation distances and the detection regions (sets of values of $(\bar{\kappa}_{\text{in-in}}, \bar{\kappa}_{\text{in-out}})$ for which it is possible to detect the feature map $\pi^*$). This is in sharp contrast with the outlier-free case, where consistent estimation requires $\bar{\kappa}$ to be at least of order $(d \log n)^{1/4}$ irrespective from the behaviour of variances of $P_i$. We prove in this work that in the high dimensional regime $d \geq c \log n$, which is arguably more appealing than the low dimensional regime $d \leq c \log n$, the following statements are true:

- If there is no heteroscedasticity, *i.e.*, when all the variances are equal, consistent estimation of $\pi^*$ is possible if and only if $\bar{\kappa} = \bar{\kappa}_{\text{in-in}} \wedge \bar{\kappa}_{\text{in-out}}$ is at least of order $(d \log(nm))^{1/4}$. This is the same rate as in the outlier-free case.

- If the heteroscedasticity is mild, *i.e.*, all the variances are of the same order, the condition $\bar{\kappa}_{\text{in-in}} \gtrsim (d \log(nm))^{1/4}$ is the same as in the previous item, but the stronger condition $\bar{\kappa}_{\text{in-out}} \gtrsim d^{1/2}$ is needed for the inlier-outlier separation distance.

- Finally, in the general heteroscedastic setting both $\bar{\kappa}_{\text{in-in}}$ and $\bar{\kappa}_{\text{in-out}}$ should be at least of order $d^{1/2}$. Furthermore, in all these cases consistent estimation is performed by the same estimator: the Least Sum of Logarithms (LSL).

Note also that the empirical evaluation done in this paper shows that LSL is interesting not only from the theoretical but also from the practical point of view.

**Agenda**   Section 2 describes the framework of the vector matching problem and introduces the terminology used throughout this paper. Precise statements of the main theoretical results are gathered in Section 3. The prior work is briefly discussed in Section 4. Section 5 contains numerical experiments carried out both for synthetic and real data. A brief summary and some concluding remarks are presented in Section 6. Proofs of all theoretical claims are deferred to the supplemental material.

## 2   Problem Formulation

We begin with formalizing the problem of matching two sequences of feature vectors $(X_1, \ldots, X_n)$ and $(X_1^{\#}, \ldots, X_m^{\#})$ with different sizes $n$ and $m$ such that $m \geq n \geq 2$. In what follows, we assume

---

[1] We use the notation $\mathbf{I}_d$ for the $d \times d$ identity matrix

that the observed feature vectors are randomly generated from the model

$$\begin{cases} X_i = \theta_i + \sigma_i \xi_i\,, \\ X_j^\# = \theta_j^\# + \sigma_j^\# \xi_j^\#, \end{cases} \qquad i = 1, \ldots, n \text{ and } j = 1, \ldots, m. \tag{1}$$

In this model, illustrated in Figure 1, it is assumed that

- $\boldsymbol{\theta} = (\theta_1, \ldots, \theta_n)$ and $\boldsymbol{\theta}^\# = (\theta_1^\#, \ldots, \theta_m^\#)$ are two sequences of vectors from $\mathbb{R}^d$, corresponding to the original features, which are unavailable,
- $\boldsymbol{\sigma} = (\sigma_1, \ldots, \sigma_n)^\top, \boldsymbol{\sigma}^\# = (\sigma_1^\#, \ldots, \sigma_m^\#)^\top$ are positive real numbers corresponding to the magnitudes of the noise contaminating each feature,
- $\xi_1, \ldots, \xi_n$ and $\xi_1^\#, \ldots, \xi_m^\#$ are two independent sequences of i.i.d. random vectors drawn from the Gaussian distribution with zero mean and identity covariance matrix.

The simplest special case of (1), considered in (Collier and Dalalyan, 2016), corresponds to the situation where a perfect matching exists between the two sequences $\boldsymbol{\theta}$ and $\boldsymbol{\theta}^\#$. This means that $m = n$ and, for some bijective mapping $\pi^* : [n] \to [n]$, $\theta_i = \theta_{\pi(i)}^\#$ for all $i \in [n]$. In the general case, both $\boldsymbol{X} = (X_1, \ldots, X_n)$ and $\boldsymbol{X}^\# = (X_1^\#, \ldots, X_m^\#)$ may contain outliers, *i.e.* feature vectors that have no corresponding pair. In such a situation, it is merely assumed that there exists a set $S \subset [n]$ and an injective mapping $\pi^* : S \to [m]$ such that

$$\theta_i = \theta_{\pi^*(i)}^\# \quad \text{and} \quad \sigma_i = \sigma_{\pi^*(i)}^\#, \qquad \forall i \in S. \tag{2}$$

In this case we say that the vectors $\{X_i : i \in [n] \setminus S\}$ and $\{X_j^\# : j \in [m] \setminus \pi^*(S)\}$ are outliers. The ultimate goal is to estimate the feature matching map $\pi^*$.

In this work we consider the case when $S = [n]$ and $m > n$. This means that only the larger set of feature vectors, namely $\boldsymbol{X}^\#$, contains outliers. Let us also define the set $O_{\pi^*} \triangleq [m] \setminus \text{Im}(\pi^*)$, which contains the indices of outliers and satisfies $|O_{\pi^*}| = m - n$. Naturally, the feature vectors contained in $\boldsymbol{X}$, as well as those vectors from $\boldsymbol{X}^\#$ that are not outliers, are called *inliers*.

In this formulation, the data generating distribution is defined by the parameters $\boldsymbol{\theta}^\#$, $\boldsymbol{\sigma}^\#$ and $\pi^*$. We omit the set of parameters $\boldsymbol{\theta}$ and $\boldsymbol{\sigma}$, since they are automatically identified using $\pi^*$, $\boldsymbol{\theta}^\#$ and $\boldsymbol{\sigma}^\#$ by the formula $(\theta_i, \sigma_i) = (\theta_{\pi^*(i)}^\#, \sigma_{\pi^*(i)}^\#)$ for $i \in [n]$. Since our goal is to match the feature vectors, we focus our attention on the problem of estimating the parameter $\pi^*$ only, considering $\boldsymbol{\theta}^\#$ and $\boldsymbol{\sigma}^\#$ as nuisance parameters. In what follows, we denote by $\mathbf{P}_{\boldsymbol{\theta}^\#, \boldsymbol{\sigma}^\#, \pi^*}$ the probability distribution of the sequence $(X_1, \ldots, X_n, X_1^\#, \ldots, X_m^\#)$ defined by (1) under condition (2) with $S = [n]$.

We are interested in designing estimators that have an expected error smaller than a prescribed level $\alpha$ under the weakest possible conditions on the nuisance parameter $\boldsymbol{\theta}^\#$ and noise level $\boldsymbol{\sigma}^\#$. Clearly, the problem of matching becomes more difficult with hardly distinguishable features. To quantify this phenomenon, we introduce the normalized separation distance $\bar{\kappa}_{\text{in-in}} = \bar{\kappa}_{\text{in-in}}(\boldsymbol{\theta}^\#, \boldsymbol{\sigma}^\#, \pi^*)$ and the normalized outlier separation distance $\bar{\kappa}_{\text{in-out}} = \bar{\kappa}_{\text{in-out}}(\boldsymbol{\theta}^\#, \boldsymbol{\sigma}^\#, \pi^*)$, which measure the minimal distance-to-noise ratio between inliers and the minimal distance-to-noise ratio between inliers and outliers, respectively. The precise definitions read as

$$\bar{\kappa}_{\text{in-in}} \triangleq \min_{\substack{i,j \notin O_{\pi^*}, \\ j \neq i}} \frac{\|\theta_i^\# - \theta_j^\#\|}{(\sigma_i^{\#2} + \sigma_j^{\#2})^{1/2}}, \qquad \bar{\kappa}_{\text{in-out}} \triangleq \min_{\substack{i \notin O_{\pi^*}, \\ j \in O_{\pi^*}}} \frac{\|\theta_i^\# - \theta_j^\#\|}{(\sigma_i^{\#2} + \sigma_j^{\#2})^{1/2}}. \tag{3}$$

Notice that $\bar{\kappa}_{\text{in-in}}$ can be rewritten as

$$\bar{\kappa}_{\text{in-in}} = \min_{\substack{i,j \in [n] \\ i \neq j}} \frac{\|\theta_i - \theta_j\|}{(\sigma_i^2 + \sigma_j^2)^{1/2}}.$$

Clearly, if $\bar{\kappa}_{\text{in-in}} = 0$ or, $\bar{\kappa}_{\text{in-out}} = 0$, there are two identical feature vectors in $\boldsymbol{X}^\#$. In such a situation, assuming $\sigma_i$'s are all equal, the parameter $\pi^*$ is nonidentifiable, in the sense that there exist two different permutations $\pi_1^*$ and $\pi_2^*$ such that the distributions $\mathbf{P}_{\boldsymbol{\theta}^\#, \boldsymbol{\sigma}^\#, \pi_1^*}$ and $\mathbf{P}_{\boldsymbol{\theta}^\#, \boldsymbol{\sigma}^\#, \pi_2^*}$ coincide. Therefore, to ensure the existence of consistent estimators of $\pi^*$ it is necessary to impose the conditions $\bar{\kappa}_{\text{in-in}} > 0$ and $\bar{\kappa}_{\text{in-out}} > 0$. Moreover, good estimators are those consistently estimating $\pi^*$ even if either $\bar{\kappa}_{\text{in-in}}$ or $\bar{\kappa}_{\text{in-out}}$ are small. We are interested here in finding the detection boundary in terms of the order of magnitude of $(\bar{\kappa}_{\text{in-in}}, \bar{\kappa}_{\text{in-out}})$. More precisely, for any given $\alpha \in (0, 1)$ we wish to find a region $\mathcal{R}_{n,m,d}^\alpha$ in $\mathbb{R}^2$ such that:

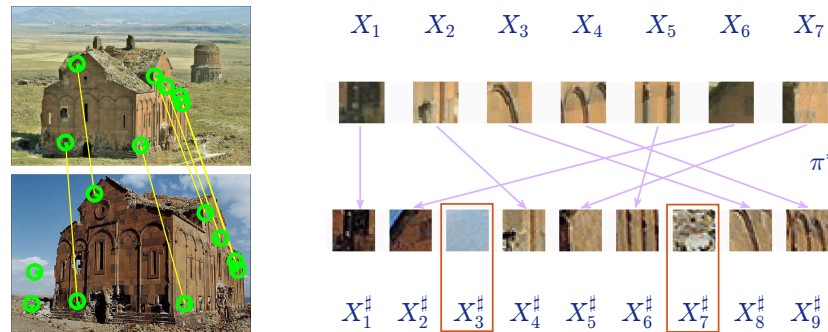

Figure 1: Illustration of the considered framework described in (1). We wish to match a set of 7 patches extracted from the first image to the 9 patches from the second image. The picture on the left shows the locations of patches as well as the true matching map $\pi^*$ (the yellow lines).

- There is an estimator $\hat{\pi}_{n,m}$ of $\pi^*$ satisfying $\mathbf{P}_{\boldsymbol{\theta}^\#,\boldsymbol{\sigma}^\#,\pi^*}(\hat{\pi} \neq \pi^*) \leq \alpha$ for every $(\boldsymbol{\theta}^\#,\boldsymbol{\sigma}^\#,\pi^*)$ lying in the detection region, *i.e.*, for which $(\bar{\kappa}_{\text{in-in}}, \bar{\kappa}_{\text{in-out}}) \in \mathcal{R}^\alpha_{n,m,d}$.

- There is a constant $C < 1$ such that for any estimator $\bar{\pi}_{n,m}$ of $\pi^*$, we can find a parameter value $(\boldsymbol{\theta}^\#,\boldsymbol{\sigma}^\#,\pi^*)$ in the region $\{(\boldsymbol{\theta}^\#,\boldsymbol{\sigma}^\#,\pi^*) : (\bar{\kappa}_{\text{in-in}}, \bar{\kappa}_{\text{in-out}}) \in C\mathcal{R}^\alpha_{n,m,d}\}$ such that $\bar{\pi}$ fails to detect $\pi^*$ with a probability larger than $\alpha$.

Let us make two remarks. First, note that in the outlier-free case considered in (Collier and Dalalyan, 2016), $\bar{\kappa}_{\text{in-out}}$ is meaningless and, therefore, the detection region is one-dimensional $\mathcal{R}^\alpha_{n,m,d}$. Thus, it is necessarily a half-line and is proven to be of the form $\bar{\kappa}_{\text{in-in}} \geq C(\log n/\alpha)^{1/2} \vee (d \log n/\alpha)^{1/4}$ for some universal constant $C$. Second, the aforementioned definition of the detection region $\mathcal{R}^\alpha_{n,m,d}$ does not guarantee its uniqueness (even up to a scaling by a universal constant). This is in contrast with the outlier-free case. To overcome this difficulty, we look for $\mathcal{R}^\alpha_{n,m,d}$ of the form $[t_{\text{in-in}}, +\infty) \times [t_{\text{in-out}}, +\infty)$ with the smallest possible threshold $t_{\text{in-out}}$ for the normalized inlier-outlier distance $\bar{\kappa}_{\text{in-out}}$.

# 3 Main theoretical results

In this section, we have collected the main theoretical findings of the paper. When the noise is homoscedastic, *i.e.*, when all $\boldsymbol{\sigma}$'s are equal, the results obtained by Collier and Dalalyan (2016) in the outlier-free setting can be easily extended to the setting with outliers. Therefore, in the present paper, we focus on the heteroscedastic case. For the sake of clarity of exposition, we will present the results in the case of known variances $\boldsymbol{\sigma}, \boldsymbol{\sigma}^\#$ prior to investigating the more interesting case of unknown variances.

The detection regions we study below are based on the profile maximum likelihood estimator. The model presented in (1) has the parameter $\boldsymbol{\Xi} = (\boldsymbol{\theta}^\#, \boldsymbol{\sigma}^\#, \pi)$, while the observations are the sequences of feature vectors $\boldsymbol{X}$ and $\boldsymbol{X}^\#$. The negative log-likelihood of this model is given by

$$\ell_n(\boldsymbol{\Xi}; \{\boldsymbol{X}, \boldsymbol{X}^\#\}) = \sum_{i=1}^n \left( \frac{\|X_i - \theta^\#_{\pi(i)}\|_2^2}{2\sigma^{\#2}_{\pi(i)}} + \frac{1}{2}\log(\sigma^{\#2}_{\pi(i)}) \right) + \sum_{j=1}^m \left( \frac{\|X^\#_j - \theta^\#_j\|_2^2}{2\sigma^{\#2}_j} + \frac{1}{2}\log(\sigma^{\#2}_j) \right).$$

The profile negative log-likelihood is then defined as the minimum with respect to $(\boldsymbol{\theta}^\#, \boldsymbol{\sigma}^\#)$ of the log-likelihood $\ell_n(\boldsymbol{\Xi}; \{\boldsymbol{X}, \boldsymbol{X}^\#\})$.

## 3.1 Warming up: known variances $\boldsymbol{\sigma}, \boldsymbol{\sigma}^\#$

One can check that the minimization with respect to $\boldsymbol{\theta}^\#$ leads to the variance-dependent cost function

$$\ell_n(\pi, \boldsymbol{\sigma}^\#; \{\boldsymbol{X}, \boldsymbol{X}^\#\}) = \sum_{i=1}^n \frac{\|X_i - X^\#_{\pi(i)}\|^2}{\sigma_i^2 + \sigma^{\#2}_{\pi(i)}} + \sum_{i=1}^n \frac{1}{2}\log(\sigma^{\#2}_{\pi(i)}) + \sum_{j=1}^m \frac{1}{2}\log(\sigma^{\#2}_j). \qquad (4)$$

When $m = n$ and there is no outlier, the last two sums of the last display do not depend on $\pi$ and, therefore, the maximum profile likelihood estimator of $\pi^*$ is obtained by the Least Sum of

Normalized Squares (LSNS) criterion

$$\hat{\pi}_{n,m}^{\mathrm{LSNS}} \in \arg\min_{\pi} \sum_{i=1}^{n} \frac{\|X_i - X_{\pi(i)}^{\#}\|^2}{\sigma_i^2 + {\sigma_{\pi(i)}^{\#}}^2}, \tag{5}$$

where the minimum is over all injective mappings $\pi : [n] \to [m]$. This, and the other estimators defined in this work, can be efficiently computed using suitable versions of the Hungarian algorithm (Kuhn, 1955, 2010; Munkres, 1957). As shows the next theorem, it turns out that even when $m > n$, the estimator $\hat{\pi}_{n,m}^{\mathrm{LSNS}}$ defined above leads to an optimal detection region.

**Theorem 1 (Upper bound for LSNS)** *Let $\alpha \in (0, 1)$ and condition* (2) *be fulfilled. If the separation distances $\bar{\kappa}_{\text{in-in}}$ and $\bar{\kappa}_{\text{in-out}}$ corresponding to $(\boldsymbol{\theta}^{\#}, \boldsymbol{\sigma}^{\#}, \pi^*)$ and defined in* (3) *satisfy the condition*

$$\min\{\bar{\kappa}_{\text{in-in}}, \bar{\kappa}_{\text{in-out}}\} \geq 4\Big\{ \big(d \log(4nm/\alpha)\big)^{1/4} \vee \big(2 \log(8nm/\alpha)\big)^{1/2} \Big\} \tag{6}$$

*then the LSNS estimator defined in* (5) *detects the true matching map $\pi^*$ with probability at least $1 - \alpha$, that is*

$$\mathbf{P}_{\boldsymbol{\theta}^{\#}, \boldsymbol{\sigma}^{\#}, \pi^*}(\hat{\pi}_{n,m}^{\mathrm{LSNS}} = \pi^*) \geq 1 - \alpha.$$

The similarity—both its statement and its proof— of this result is to its counterpart in the outlier-free setting might suggest that the presence of outliers does not make the problem any harder from a statistical point of view. However, this is not true in the more appealing setting of unknown variances.

## 3.2 Detection of $\pi^*$ for unknown and arbitrary variances $\boldsymbol{\sigma}, \boldsymbol{\sigma}^{\#}$

The LSNS procedure analyzed in Theorem 1 exploits the values of known noise variances to normalize the squares of distances between vectors $X_i$ and $X_{\pi(i)}^{\#}$. Therefore, LSNS is inapplicable in the case of unknown noise variances. Instead, we consider the Least Sum of Logarithms (LSL) estimator

$$\hat{\pi}_{n,m}^{\mathrm{LSL}} \triangleq \arg\min_{\pi:[n]\to[m]} \sum_{i=1}^{n} \log \|X_i - X_{\pi(i)}^{\#}\|^2, \tag{7}$$

where the minimum is over all injective maps $\pi : [n] \to [m]$. This estimator can be seen as the minimizer of a criterion defined as the minimum of the cost function from (4) with respect to $\boldsymbol{\sigma}^{\#}$ under the constraint $\min_{j \notin \mathrm{Im}(\pi)} \sigma_j^{\#} \geq \sigma_{\min}$, for some fixed (but unknown) constant $\sigma_{\min} > 0$.

To provide a quick overview of what follows, let us stick in the remaining of this paragraph to the case $\log(nm) = O(d)$ so that the right hand side of (6) is of order $\big(d \log(nm)\big)^{1/4}$. Recall that in the outlier-free case, the LSL estimator has been shown to perform as well as the LSNS while having the advantage of not requiring the knowledge of variances $\boldsymbol{\sigma}^{\#}$ (Collier and Dalalyan, 2016). Somewhat unexpectedly, the situation is significantly different in the presence of outliers. Indeed, the best we manage to prove in the presence of outliers is that the detection of the matching map by LSL is possible whenever $\min\{\bar{\kappa}_{\text{in-in}}, \bar{\kappa}_{\text{in-out}}\} \geq C\sqrt{d}$ for some sufficiently large constant $C$. The precise statement being given in the next theorem, let us mention right away that the discrepancy between this rate $\sqrt{d}$ and the rate $\big(d \log(nm)\big)^{1/4}$ in (6) is due to the inherent hardness of the setting and not merely an artefact of the proof. This will be made clear below.

**Theorem 2 (Upper bound for LSL)** *Let $\alpha \in (0, 1/2)$ and condition* (2) *be fulfilled. If the separation distances $\bar{\kappa}_{\text{in-in}}$ and $\bar{\kappa}_{\text{in-out}}$ corresponding to $(\boldsymbol{\theta}^{\#}, \boldsymbol{\sigma}^{\#}, \pi^*)$ and defined by* (3) *satisfy*

$$\min\{\bar{\kappa}_{\text{in-in}}, \bar{\kappa}_{\text{in-out}}\} \geq \sqrt{2d} + 4\Big\{ \big(2d \log(4nm/\alpha)\big)^{1/4} \vee \big(3 \log(8nm/\alpha)\big)^{1/2} \Big\} \tag{8}$$

*then the LSL estimator* (7) *detects the matching map $\pi^*$ with probability at least $1 - \alpha$, that is*

$$\mathbf{P}_{\boldsymbol{\theta}^{\#}, \boldsymbol{\sigma}^{\#}, \pi^*}(\hat{\pi}_{n,m}^{\mathrm{LSL}} = \pi^*) \geq 1 - \alpha.$$

This result is disappointing since it requires the distance between different feature vectors to be larger than $\sqrt{2d}$ in order to be able to consistently estimate the matching map $\pi^*$. As we show below, without any further condition (for instance, on the noise variances), this rate cannot be improved.

Furthermore, the rate $\sqrt{d}$ is optimal not only for LSL but also for the larger class of so called *distance based M-estimators.*

We will say that an estimator $\hat{\pi}_n$ of $\pi^*$ is a distance based $M$-estimator, if for a sequence of non-decreasing functions $\rho_i : \mathbb{R}_+ \to \mathbb{R}$, $i = 1, \ldots, n$, the following is correct

$$\hat{\pi}_n \in \arg\min_{\pi:[n]\to[m]} \sum_{i=1}^n \rho_i\big(\|X_i - X_{\pi(i)}^{\#}\|\big),$$

where the minimum is over all injective mappings $\pi : [n] \to [m]$. We denote by $\mathcal{M}$ the set of all distance based $M$-estimators. We show that there is indeed a setup where $\bar{\kappa}_{\text{in-in}} \wedge \bar{\kappa}_{\text{in-out}}$ is as large as $0.2\sqrt{d}$ but any estimator from $\mathcal{M}$ fails to detect $\pi^*$ with probability at least $1/4$. The next theorem formalizes the described result.

**Theorem 3 (Lower bound)** *Assume that $m > n \geq 4$ and $d \geq 422\log(4n)$. There exists a triplet $(\boldsymbol{\sigma}^{\#}, \boldsymbol{\theta}^{\#}, \pi^*)$ such that $\bar{\kappa}_{\text{in-in}} = \bar{\kappa}_{\text{in-out}} = \sqrt{d/20}$ and*

$$\inf_{\hat{\pi}\in\mathcal{M}} \mathbf{P}_{\boldsymbol{\theta}^{\#},\boldsymbol{\sigma}^{\#},\pi^*}(\hat{\pi} \neq \pi^*) > 1/4. \tag{9}$$

The proof of this theorem, postponed to the appendix, is constructive. This means that we exhibit a triplet $(\boldsymbol{\sigma}^{\#}, \boldsymbol{\theta}^{\#}, \pi^*)$ satisfying (9). Careful inspection shows that in the case $d = O(\log(nm))$ the same triplet satisfies $\bar{\kappa}_{\text{in-in}} \wedge \bar{\kappa}_{\text{in-out}} \asymp \sqrt{\log(nm)}$ and (9) is still true. This implies that the order of magnitude of the right hand side of (8) is optimal both in the high-dimensional regime $d \geq 422\log(4n)$ and in the low-dimensional regime $d < 422\log(4n)$. This shows the optimality of LSL among all estimators from $\mathcal{M}$. Note that the estimator $\hat{\pi}_{n,m}^{\text{LSNS}}$ does not belong to the family of *distance based M-estimators.* Furthermore, in the low dimensional regime $d = O(\log(nm))$, the separation rate of the LSL, $\sqrt{\log(nm)}$, is the same as that of the LSNS.

In the next section we show that under some mild conditions on $\boldsymbol{\sigma}^{\#}$ it is indeed possible to obtain different rates for $\bar{\kappa}_{\text{in-in}}$ and $\bar{\kappa}_{\text{in-out}}$, namely we show that if $\bar{\kappa}_{\text{in-in}} \gtrsim d^{1/4}$ and $\bar{\kappa}_{\text{in-out}} \gtrsim d^{1/2}$ then the LSL estimator detects correct matching with high probability.

## 3.3 Detection of $\pi^*$ for unknown and mildly varying variances $\boldsymbol{\sigma}, \boldsymbol{\sigma}^{\#}$

The results of the last two theorems are disappointing, since they indicate that the features should be very different from one another for detection of the matching map to be possible. An interesting finding, presented below, is that strong constraint can be significantly alleviated in the context of mild heteroscedasticity. By mild heteroscedasticity we understand here the situation in which all variances $\sigma_i^{\#}$ are of the same order of magnitude.

**Theorem 4 (Upper bound under mild heteroscedasticity)** *Let $r_\sigma = \max_{i,j}(\sigma_i^{\#}/\sigma_j^{\#}) < \infty$. If the separation distances $\bar{\kappa}_{\text{in-in}}$ and $\bar{\kappa}_{\text{in-out}}$ defined in (3) satisfy*

$$\bar{\kappa}_{\text{in-in}} \geq 2\big(4d\log(4nm/\alpha)\big)^{1/4} + 2\big(2\log(4nm/\alpha)\big)^{1/2}$$
$$\bar{\kappa}_{\text{in-out}} \geq \sqrt{2(r_\sigma-1)d} + 2\big(4r_\sigma^2 d\log(4nm/\alpha)\big)^{1/4} + 2\big(2r_\sigma\log(4nm/\alpha)\big)^{1/2},$$

*then the LSL estimator (7) detects the matching map $\pi^*$ with probability at least $1 - \alpha$, that is*

$$\mathbf{P}_{\boldsymbol{\theta}^{\#},\boldsymbol{\sigma}^{\#},\pi^*}(\hat{\pi}_{n,m}^{\text{LSL}} = \pi^*) \geq 1 - \alpha.$$

Note that a lower bound similar to that of Theorem 3 can be proved in the case of mild heterescodestacity as well, showing that there is an example for which $\bar{\kappa}_{\text{in-in}}$ is of order $d^{1/4}$, $\bar{\kappa}_{\text{in-out}}$ is of order $d^{1/2}$ and any estimator from $\mathcal{M}$ fails to detect $\pi^*$ with probability at least $1/4$.

We complete this section by summarizing the joint contribution of Theorems 1 to 4. To simplify this discussion, we consider two cases: high-dimensional case refers to $d \geq \log(4nm/\alpha)$ (presented in Table 1) and low-dimensional case refers to the condition $d < \log(4nm/\alpha)$. In the high dimensional setting with arbitrary noise variances, the detection region for the LSL estimator is given by $\{\bar{\kappa}_{\text{in-in}} \wedge \bar{\kappa}_{\text{in-out}} \geq 15\sqrt{d}\}$, which is much worse than the detection region for LSNS, $\{\bar{\kappa}_{\text{in-in}} \wedge \bar{\kappa}_{\text{in-out}} \geq 8(d\log(4nm/\alpha))^{1/4}\}$, obtained in the known-variance scenario. Somewhat surprisingly, in such a

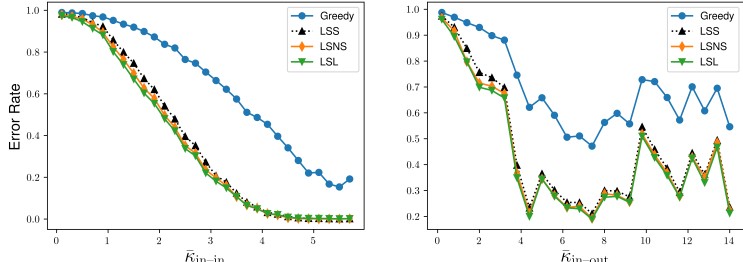

Figure 2: The performance of the methods (Greedy, LSS, LSL, LSNS) in the setup described in Exp. 1. Plots show the error rate (percentage of repetitions in which the estimated matching map differs from the true one) as a function of separation distances. The left plot illustrates that LSS, LSL and LSNS require much lower value of $\bar{\kappa}_{\text{in-in}}$ in order to find the correct mapping. The left plot shows that there is a clear improvement of the error when the inlier-inlier separation distance increases, while the right plot shows that the error rate might not be made small by augmenting $\bar{\kappa}_{\text{in-out}}$ only.

setting, even a strong assumption on the outliers, such as requiring them to be at least at a distance $0.2\sqrt{d}$ of the inliers, is not enough for relaxing the assumption on the inlier-inlier separation distance. Finally, on a positive note, in the intermediate case of mildly varying variances, the detection region for the LSL estimator is of the form $\{\bar{\kappa}_{\text{in-in}} \geq 7(d \log(4nm/\alpha))^{1/4};\ \bar{\kappa}_{\text{in-out}} \geq 10\sqrt{d}\}$. This means that if the outliers are at a distance $\Omega(\sqrt{d})$ of the inliers, then the LSL recovers the true matching under the same condition on $\bar{\kappa}_{\text{in-in}}$ as in the outlier-free setting.

## 4  Other related work

Measuring the quality of the various statistical procedures of decision making by their minimal separation rates became the standard in hypotheses testing, see the seminal papers (Burnashev, 1979; Ingster, 1982) and the monographs (Ingster and Suslina, 2003; Juditsky and Nemirovski, 2020). Currently this approach is widely adopted in machine learning literature (Xing et al., 2020; Wolfer and Kontorovich, 2020; Blanchard et al., 2018; Ramdas et al., 2016; Wei et al., 2019; Collier, 2012). Beyond the classical setting of two hypotheses, it can also be applied to multiple hypotheses testing framework, for instance, variable selection (Ndaoud and Tsybakov, 2020; Azaïs and de Castro, 2020; Comminges and Dalalyan, 2012) or the matching problem considered here.

On the other hand, feature matching is a well studied problem in computer vision. In recent years, a great deal of attention was devoted to the acceleration of greedy matching algorithms, based on approximate and fast methods of finding nearest neighbors (e.g. Jiang et al. (2016); Wang et al. (2018); Wang (2011); Harwood and Drummond (2016); Malkov and Yashunin (2020)). Another direction that helps to improve feature matching problem is using alternative local descriptors (Rublee et al., 2011; Chen et al., 2010; Calonder et al., 2010) for given keypoints. Naturally, the question of how to chose keypoints arises, which is addressed, for instance, in (Bai et al., 2020; Tian et al., 2020). For more complete overview of the field we refer to (Ma et al., 2021) and references therein.

Finally, permutation estimation and related problems have been recently investigated in different contexts such as statistical seriation (Flammarion et al., 2019), noisy sorting (Mao et al., 2018), regression with shuffled data (Pananjady et al., 2017; Slawski and Ben-David, 2019), isotonic regression and matrices (Mao et al., 2020; Pananjady and Samworth, 2020; Ma et al., 2020), crowd labeling (Shah et al., 2021), and recovery of general discrete structure (Gao and Zhang, 2019).

## 5  Numerical results

In this section, we report the results of some numerical experiments carried out on simulated and real data. We applied aforementioned methods LSNS and LSL and computed different measures of their performance. To get a more complete picture, we included in this study the Least Sum of Squeres (LSS) estimator and the greedy estimator. LSS is an unnormalized version of LSNS, given by

$$\hat{\pi}_{n,m}^{\text{LSS}} \in \arg\min_{\pi} \sum_{i=1}^{n} \|X_i - X_{\pi(i)}^{\#}\|^2. \tag{10}$$

It coincides with LSNS in the case of homoscedastaic noise. The greedy estimator is obtained by sequentially matching each vector from $\boldsymbol{X}$ to the nearest vector from $\boldsymbol{X}^{\#}$. Experiments were implemented using python or matlab. For solving linear sum assignment problems such as (7) or (10), the generalized and improved versions of the Hungarian algorithm were used (Kuhn, 1955, 2010; Munkres, 1957; Duff and Koster, 2001), implemented in SciPy library (Virtanen et al., 2020).

**Experiment 1: Synthetic data with random features** We first randomly generated $\pi^*$, $\boldsymbol{\theta}^{\#}$ and $\boldsymbol{\sigma}^{\#}$ as follows. We randomly sampled from uniform distribution on $[0, 2]$ independent variables $\tau_{ij}$, $i \in [m], j \in [d]$. Then, $(\theta_i^{\#})_j$ are independently sampled from the Gaussian distribution with 0 mean and variance $\tau_{ij}$. Additionally, for every $\theta_i^{\#} \in \boldsymbol{\theta}^{\#}$ such that $i \notin J_{\pi^*}$ ($\theta_i^{\#}$ is an outlier), we incremented every coordinate of $\theta_i^{\#}$ by $i$. Entries of $\boldsymbol{\sigma}^{\#}$ were sampled from the uniform distribution over $[0.5, 2]$. Sequences $\boldsymbol{X}$ and $\boldsymbol{X}^{\#}$ were generated according to Section 2 with $\pi^*(i) = i$ for $i \in [n]$. We applied to this data the following matching algorithms: Greedy, LSS, LSNS and LSL.

We chose $n = 100$, $m = 130$ and $d = 50$, and generated $N = 50$ datasets according to the foregoing process. For each dataset, we computed the 0-1 error of the considered estimators and the values of $(\bar{\kappa}_{\text{in-in}}, \bar{\kappa}_{\text{in-out}})$. We plotted in the left (resp., the right) panel of Figure 2 the error averaged over all datasets with a given value of $\bar{\kappa}_{\text{in-in}}$ (resp. $\bar{\kappa}_{\text{in-out}}$). In this specific example, we see that the error decreases fast with $\bar{\kappa}_{\text{in-in}}$, while there is no monotonicity of the error rate as a function of $\bar{\kappa}_{\text{in-out}}$.

**Experiment 2: Synthetic data with deterministic features** The second experiment is conducted on data generated by features $\boldsymbol{\theta}^{\#}$ and variances $\boldsymbol{\sigma}^{\#}$ inspired by the example constructed in the proof of Theorem 3. More precisely, for some real numbers $a$ and $b$ representing, respectively, the scale of inlier-inlier distance $\bar{\kappa}_{\text{in-in}}$ and inlier-outlier distance $\bar{\kappa}_{\text{in-out}}$, we set $\theta_k^{\#} = [ka, 0, \ldots, 0]^{\top}$ for $k \in [n]$ and $\theta_{n+k}^{\#} = [na + kb, 0, \ldots, 0]^{\top}$. We also used decreasing variances $\sigma_k^{\#} = 1/k^{3/2}$ for $k \in [m]$ and true identity mapping $\pi^*(k) = k$ for $k \in [n]$. We chose $n = 100$, $m = 120$ and dimension $d$ varying in the set $\{10, 20, 40\}$. For each pair of values $(a, b)$ in a suitably chosen grid, we repeated $n_{\text{rep}} = 400$ times the experiment that consisted in generating data according to (1) and computing estimators $\hat{\pi}_{n,m}^{\text{LSS}}$ and $\hat{\pi}_{n,m}^{\text{LSL}}$ defined respectively by (10) and (7). We then computed, for each pair $(a, b)$ and for each estimator LSS and LSL, the percentage of successful detection among $n_{\text{rep}}$ repetitions.

The obtained detection regions are depicted in Figure 3 in the form of heatmaps. This visualisation allows us to grasp the forms of the detection regions for the specific choice of parameters considered in this example. The first observation is that LSL is clearly superior to LSS for all the considered values of the dimension. Second, we clearly see the deterioration of the detection region when the dimension $d$ becomes larger. Third, the values of $\bar{\kappa}_{\text{in-out}}$ used in the plots are at least one order of magnitude larger than those of $\bar{\kappa}_{\text{in-in}}$. This is in line with the claim of Theorem 4. We also observe in these pictures that successful detection occurs when $\bar{\kappa}_{\text{in-out}}$ is larger than some threshold even if $\bar{\kappa}_{\text{in-in}}$ is small. This must be a nice feature of LSL and LSS in this specific example, which unfortunately does not generalize to other examples as shown by our theoretical results.

**Experiment 3: Real data example** This experiment is conducted on the IMC-PT 2020 dataset from Jin et al. (2020) that consists of images of 16 different scenes with corresponding 3D point-clouds of landmarks, which are used to obtain (pseudo) ground-truth local keypoint matchings. For a given scene, we sampled 1000 pairs of distinct images of the same landmark with different camera locations, angles, weather conditions etc. For each image pair we generated 2D keypoints from original set of 3D points (note that the same 3D point appears in both of the images, so we have ground truth keypoint matching between 2 images). Subsequently, we computed SIFT descriptors (Lowe, 2004) for every keypoint in images using Python OpenCV interface (Itseez, 2015). Some pairs of images being more challenging than others, we split the dataset into two sets of image pairs in order to gain more understanding on the behaviour of the algorithms. The challenging pairs are those for which the OpenCV default matching algorithm has accuracy less than $0.5$. Then, for every image pair, we fixed randomly chosen 100 keypoints in the first image (and corresponding keypoints in the second image) and added outliers to the second image from the remaining keypoints. The outlier rate is chosen to be between $0\%$ to $70\%$. Finally, 3 descriptor matching algorithms were applied (OpenCV default matching algorithm, LSS and LSL). Note that $\boldsymbol{\sigma}$ and $\boldsymbol{\sigma}^{\#}$ from (1) are unknown, hence LSNS is not applicable. One can consider using the estimates $\hat{\boldsymbol{\sigma}}$ and $\hat{\boldsymbol{\sigma}}^{\#}$ instead of $\boldsymbol{\sigma}$ and $\boldsymbol{\sigma}^{\#}$ in (5), but this is beyond the scope of this paper.. Further details on this experiment along with some additional results are deferred to the supplemental material (Appendix D).

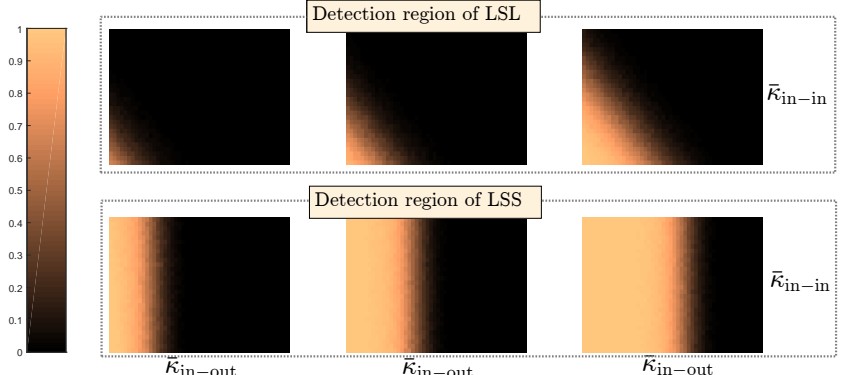

Figure 3: Heatmaps of the error rate of the LSL (top row) and the LSS (bottom row) estimators in Experiment 2. We chose $n = 100$, $m = 120$ and $d \in \{10, 20, 40\}$ from left to right. The parameter $a$ representing the scale of $\bar{\kappa}_{\text{in-in}}$ and corresponding to $y$-axis varies from 0.02 to 0.08, whereas $b$ representing the scale of $\bar{\kappa}_{\text{in-out}}$ and corresponding to $x$-axis varies from 0.3 to 10. Dark colour means that probability of successful detection is close to 1 (error rate close to zero).

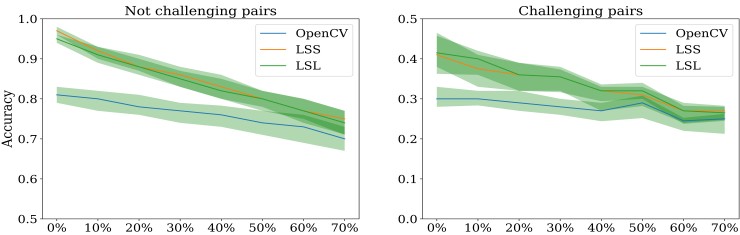

Figure 4: The estimation accuracy measured in the Hamming loss of the estimated matching in Exp. 3 for different values of the outlier rate, $(m-n)/n$, varying from $0\%$ to $70\%$. The medians of estimation accuracy both for challenging pairs (right plot) and simple pairs (left plot) of images from Temple Nara scene was computed using OpenCV, LSS and LSL matchers. The green region represents the interquartile range (lower and upper bounds being $25\%$ and $75\%$ percentiles, respectively).

The median estimation accuracy measured in the Hamming loss—for the image pairs from Temple Nara Japan scene—is plotted in Figure 4. Similar results for other scenes are presented in the supplemental material. The error bars with borderlines corresponding to $75\%$ and $25\%$ percentiles are also displayed. The first observation is that LSS and LSL outperform the OpenCV matcher in terms of Hamming distance. Second, the task of feature matching becomes harder with the growth of outlier rate, with a deterioration that seems to be linear in the rate of outliers. Notice that in this experiment the outliers can be very similar to the inliers and the separation condition imposed on $\bar{\kappa}_{\text{in-out}}$, of Theorems 2 and 4, is violated. This implies that the larger the outlier rate the harder it is to find the correct match, causing a larger number of mistakes.

# 6  Conclusion

We have investigated the detection regions in the problem of estimation of the matching map between two sequences of noisy vectors. We have shown that the presence of outliers in one of the two sequences has a strong negative impact on the detection region. Interestingly, this negative impact is mitigated in the regime of mild heteroscedasticity, *i.e.*, when noise variances are of the same order of magnitude. In the extremely favorable case of homoscedastic noise (all variances are equal), the presence of outliers does not make the problem any harder, provided that the outliers are at least as different from inliers as two distinct inliers are different one another. Precise forms of the detection region in these different cases can be found in Table 1. The results of the numerical experiments conducted on both synthetic and real data confirm our findings and, furthermore, show the good behaviour of the LSL estimator in terms of its robustness to noise and to outliers, not only in the problem of detection but also in the problem of estimation. In the future, we plan to investigate the case when both sequences contain outliers and to obtain theoretical guarantees on the estimation error measured by the Hamming distance.

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
