# OpenReview forum: "Optimal detection of the feature matching map in presence of noise and outliers"
_NeurIPS.cc/2021/Conference — NeurIPS 2021 Submitted_

### Official Review · Reviewer_KyJs · 2021-07-13

**Rating:** 6
**Confidence:** 4

**Summary:**

This paper considers the problem of matching two sets of vectors where the second set consists of a noisy version of the first set (inliers) together with outliers. Under a statistical framework, the paper develops theoretical bounds on the inlier-inlier and inlier-outlier separation that enable correct matching for a class of estimators including the Least-Sum-of-Logarithms (LSL). The approach follows Collier and Dalalyan, 2016, who considered the same problem without outliers. The LSL estimator is also evaluated using synthetic data, as well as in a semi-synthetic computer vision experiment using the IMC-PT-2020 real dataset for the task of feature matching towards obtaining point correspondences from two views.

**Ethical Concerns:**

No issues as far as I can see.

**Limitations And Societal Impact:**

No issues as far as I can see.

**Main Review:**

The paper considers an interesting problem with many applications. It is very clearly written and the theoretical results appear to be original and have significance. The experiments show that LSL is an interesting estimator and are on par with the theory in the paper.

On the other hand, the paper would have been stronger if the authors had given results about the more general and practically relevant case, where both sets contain outliers. How much more difficult is that case?
Moreover, the authors could have motivated the problem more from a general machine learning point of view. That is, they could have discussed several different machine learning applications. Instead, the application aspect of the paper including the real data experiment comes from computer vision. While this is not necessarily a drawback, there is a huge literature in computer vision regarding feature matching and from that point of view, the experiment is more of a proof of concept than anything else (e.g., one would have to also consider algorithm complexity, running times, parameter tuning e.t.c. to actually conclude that LSL outperforms OpenCV in this experiment). In my opinion it would have been more interesting to investigate a less explored and upcoming application, such as record linkage (e.g., see the work of Slawski and Ben-David already cited in the submission).

Overall, this is a very well written theoretical paper, with good results, but also with plenty of room for improvement.

********** POST-REBUTTAL *****************
I thank the authors for their response, and I am glad my comments were well-taken.




**Time Spent Reviewing:**

9

---

> ### Author Response · Authors · 2021-08-09
> **Thank you for your constructive remarks**
>
> We would like to thank the reviewer for the time spent reading and evaluating our manuscript.We also thank the reviewer for a fair account of our contributions and constructive criticism. The suggestions made by the reviewer will be taken into account in the revised version of the paper (for NeurIPS, if the paper is accepted, or for a future submission). Let us very quickly comment on some points raised by the reviewer.
> 1. The case where there might be outliers in both sets of feature vectors is qualitatively different from the case considered in the present work. Without diving too much into technical details, we would simply like to point to the fact that it is not even straightforward to define an extension of the LSL and LSS in the general case. Indeed, if we minimize the sum of squared errors with respect to both the sets of outliers and the mapping between the inliers, the result will be the trivial map that considers all the vectors as outliers. Therefore, one has to somehow constrain the number of outliers, in order to circumvent this drawback. One way of doing this is to assume that we are given an upper bound on the number of outliers and to incorporate this knowledge into the estimator. Another, more appealing approach is to consider the number of outliers as a tuning parameter and to perform a kind of “model selection” with respect to this parameter. We are currently working on these extensions and expect to make our results public in the near future. We find it useful here to underline that the lessons we learned from the case considered in the paper under review are extremely useful for treating the general case.
> 2. We agree that there are many different ways to motivate the problem of matching considered in our paper. We were probably too excited by the mathematical aspects of our findings and paid insufficient attention to the motivations. The constraint on the number of pages was also one of the reasons for reducing the motivation part in favor of presenting mathematical results as clearly as possible. We will do our best for repairing this shortcoming in the revised version of the paper.

---

> > ### Comment · Reviewer_KyJs · 2021-08-31
> > **rebuttal acknowledgment**
> >
> > I thank the authors for their comments.

---

### Official Review · Reviewer_AuhE · 2021-07-17

**Rating:** 4
**Confidence:** 3

**Summary:**

This paper studies the problem of finding matches between two observations, one of which has outliers. It provides theoretical analysis about the conditions when using the Least Sum of Logarithms estimator can lead to optical solution. It studies different situation with regards to different cases of observations, providing the upper bound and lower bound theoretically. Experiments are conducted on synthetic datasets and one real image matching dataset.

**Main Review:**

Strengths
+: The paper gives theoretical analysis to an important problem in computer vision.
+: Numerical results validate the theoretical results.

Weaknesses
-: While the theoretical studies should be encouraged, the present one seems to be less influentially in practice. The assumptions make the obtained results can not be a good guide to practical image matching problem, especially the challenging ones.
-: Some obtained results from the theoretical analysis are well known ones, thus making the paper have less practical merits. For example, L312 “This implies that the larger the outlier rate the harder it is to find the correct match” L316 “We have shown that the presence of outliers in one of the two sequences has a strong negative impact on the detection region.”
-: From the given theoretical results, it seems that the image matching problem should be better solved by optimizing the Least Sum of Logarithms estimator, instead of the widely used Least Sum of Euclidean distances. However, it does not the advantages of using LSL in real data experiments. In real data experiments, it compared with an OpenCV matcher, which is the bruteforce NN matcher based on Euclidean distances. For a fair comparison, the compared matcher should also be the Hungarian algorithm, because the better matching results can be caused by such advanced matching algorithm instead of the simplest NN matcher.
-: Eq.(2) is not true in practice, especially for feature matching. In feature matching, although the matched features have identical original features \theta, the observed features are most likely to be with different variance, i.e., \sigma_i is not equal to \sigma^{#}_{\pi(i)} as shown in the latter of Eq.(2). Therefore, the theoritical results obtained by the paper is meaningless, or less useful in practice.
-: The studied case described in L161 log(nm) = O(d) does not usually hold for keypoint matching problems.

Other comments:
1. Why use the Hamming loss to evaluate the matching performance in image matching problem? What is the definition of this loss?
2. In supplement material, it would be better also provide the image matching results by the OpenCV matcher.


============= post rebuttal =======================

I appreciate the authors' feedbacks for my concerns, however, I didn't find them helpful.

(1) After reading the reply of 1 and other reviews, I still can not get the main contributions having practical influence.
About the reply of 2, I have clearly stated in the previous concerns that the real image matching experiments, the competitive method is the OpenCV matcher which is based on NN, not the LSS with Hungarian algorithm. "In real data experiments, it compared with an OpenCV matcher, which is the bruteforce NN matcher based on Euclidean distances. For a fair comparison, the compared matcher should also be the Hungarian algorithm, because the better matching results can be caused by such advanced matching algorithm instead of the simplest NN matcher." -- This is the original comment.

(2) The reply confuses my inital comment. For this point, I clearly pointed out that it is the variance being different, and of course, I mean the noise in descriptor. Please let me give an example, for image of RGB and NIR images, the descriptors of a same 3D point in the RGB images have a different variance to those of the same 3D point in the NIR images. This could also be the case of matching RGB images with different imaging conditions, such as by different cameras, different illuminations, etc.

(3) For the case log(nm) = O(d), as the authors' given example shows, 8log(10) < 20 is much less than 128, about one order of magnitude smaller. Meanwhile, in many cases, the number of keypoints is about 2k, for this case, this does not hold. In addition, even in the authors' real experiments, they used 100 vectors per pair (see also the reply to Reviewer YuHQ), which apparently violates such conditions. Is this experiment non-meaningful? or there is no way to support the theoretical results by real image matching experiments?

To sum up, the feedbacks did not address my comments, I downgrade the initial score to reject.

**Time Spent Reviewing:**

8

---

> ### Author Response · Authors · 2021-08-09
> **Authors’ answers to the questions and concerns raised by the reviewer**
>
> We would like to thank the reviewer for the time spent reading and evaluating our manuscript. We answer below the questions raised in the review and point to some inaccuracies in reviewers’ statements.
> 1. The reviewer claims that “Some obtained results from the theoretical analysis are well known ones” and provides as examples the following two sentences
> a) “This implies that the larger the outlier rate the harder it is to find the correct match”
> b)  “We have shown that the presence of outliers in one of the two sequences has a strong negative impact on the detection region.”
> extracted from our manuscript. We insist on the fact that the sentence in a) is not presented as a finding of our investigation. It is merely a small remark concerning the outcome of one experiment (a kind of sanity check). Concerning sentence b), it should not be separated from the sentence which follows this one in the text. Indeed, we have shown that the impact of the outliers is strong when the noise is severely heterogeneous, but that this impact is mitigated in the case of mildly heterogeneous noise. To the best of our knowledge, this phenomenon was not known before.
> 2. The reviewer suggests that for a fair comparison in our experiments, “the compared matcher should also be the Hungarian algorithm because the better matching results can be caused by such advanced matching algorithm instead of the simplest NN matcher.” In fact, we are not sure to well understand this remark, since both LSS and LSL included in our experimental evaluation use the Hungarian algorithm. We believe, therefore, that our experiments provide a fair comparison of the most relevant algorithms.
> 3. We find the reviewer’s claim that “Eq.(2) is not true in practice, especially for feature matching” very puzzling. Furthermore, the conclusion that “the theoritical results obtained by the paper is meaningless, or less useful in practice” appears to be too harsh. We are quite sure that our assumptions are realistic; they are perhaps the most reasonable assumptions for conducting a theoretical investigation. It is possible that the reviewer confuses the noise in the image with the noise in the local descriptor. We might have the projection of the same 3D point in 10 different images. These images might be noisy with different magnitudes of noise. We then extract 10 feature vectors corresponding to the projection of the given 3D point in these 10 images. Our assumption is that these 10 vectors are drawn from a Gaussian distribution with mean $\theta$ and variance $\sigma^2$. Does this make sense? Is there any work showing that this assumption is violated in practice?
> 4. We disagree that the case $\log(nm) = O(d)$ does not usually hold for keypoint matching problems. The feature vectors considered in computer vision are often of dimension $d = 128$ or $d = 256$. The number of feature vectors in a pair of images is often less than $10^4$. Therefore, $log(nm) \le 8\log(10)\le 20 \le d$.
> 5. The Hamming loss is the number of incorrect matches. In other terms, if $\pi$ and $\pi’$ are two matching maps, the Hamming distance between these two mappings is the number of points $i$ such that $\pi(i)\neq \pi’(i)$.
> Assessing the Hamming distance is crucial in computer vision. Indeed, once the matching is done, the goal is often to estimate the transformation related to two images of the same scene. The precision of the estimation is better if the number of incorrect matches is smaller. This is especially true for RANSAC and its variants, for which the number of randomly drawn minimal samples is determined as a function of incorrect matches. If there is no incorrect match, i.e., if the Hamming distance is zero, then one sample is enough.
> 6. The results reported in the supplementary material do contain those obtained by the OpenCV matcher, see Fig. 5. If the reviewer’s suggestion concerns the equivalent of Fig. 7 for the OpenCV matcher, we will be happy to include it in the revised version of the paper.

---

> ### Author Response · Authors · 2021-09-24
> **Reply to post-rebuttal comments**
>
> We thank the reviewer for adding post-rebuttal comments to the initial review. Unfortunately, the system does not send notifications when a review is updated, so we discovered by chance that modifications were made. There are still some misunderstandings that we would like to clarify below. In the next items, we use the same enumeration as in the post-rebuttal comments made by the reviewer.
>
> 1) Comparing with the Hungarian algorithm: The review suggests to include the Hungarian algorithm in the experimental evaluation. Our answer is that the Hungarian algorithm was included in the experiments from the beginning.
> 2) We agree, that there are some practical cases, e.g. images taken under very different conditions, where our condition (2) might be violated. But does this imply that "Equation (2) is not true in practice"?  There is a large number of practical problems of keypoint matching in which images are taken with the same camera, or cameras with similar parameters, under similar weather conditions. In all these cases our conditions appear to be perfectly realistic.
> 3) There is a real misunderstanding regarding this point. Imagine that the number of vectors is 2K, i.e. 2,000, or even $2K*1,000 = 2\times 10^6$, this means that $n$ and $m$ are both $\le 2\times 10^6$. Consequently, $\log(nm) \le \log(10^{13}) \le 30$ is much smaller than $d$. Where is the problem?

---

> > ### Comment · Reviewer_AuhE · 2021-09-24
> > **reply to the post-rebuttal feedback**
> >
> > 1. Comparing with the Hungarian algorithm. "Our answer is that the Hungarian algorithm was included in the experiments from the beginning."  --- For the real image matching matching, it compares OpenCV NN matcher, as L301 stated. If the authors meant that the results of LSS is euclidean distance with Hungarian algorithm, then the results in Fig.4 do not show advantages of the proposed LSL.
> >
> > 2. "There is a large number of practical problems of keypoint matching in which images are taken with the same camera, or cameras with similar parameters, under similar weather conditions." -- in these cases, the outlier ratio is usually not high with modern methods for local feature description/matching and many existing methods well solve this problem. What is the merit of the proposed method?
> >
> > 3. In the supplied example, the reviewer meant that "<= 20" can not be regarded as "O(128 or 256)". O(128) should be around 100, right? almost one magnitude of order larger.
> > In the given real experiments of this paper, the number of keypints is 100, then log(nm) = 4*log(10) <= 10. Saying this equals to log(nm) = O(d) (d=128 as the experiments used SIFT), this is unconvincing to the reviewerr.

---

> > > ### Author Response · Authors · 2021-09-24
> > > **our clarifications**
> > >
> > > 1) Yes, LSS is the Hungarian algorithm on the Euclidean distances. The experiments show how much LSS and LSL do improve upon the greedy matchers. They also confirm the result of Theorem 1, which says that in the case of equal variances (which entails that LSNS = LSS) the LSS is optimal.
> > > 2) The goal of the paper is *not* to propose a new method and to demonstrate that it improves on the competitors. The goal of the paper is to understand what the limits in the keypoint matching problem, in which cases this problem can be solved and in which cases it cannot be solved. We provide such conditions (cf Eq 8), prove that (a) if these conditions are violated then it is impossible to estimate the feature matching map and (b) if they are true then LSL (and in some cases LSS) succeeds in finding the good matching.
> > > 3) All right, we see now where the misunderstanding comes from. Notation $O(.)$ stands for of "equal or smaller order", not for "of the same order". The latter is denoted by $\Theta(.)$. Therefore, the condition we use on line 161 holds for all the problems the reviewer has in mind.
> > >
> > > We wish once again to thank the reviewer for the time spent to answer our questions.

---

### Official Review · Reviewer_YuHQ · 2021-07-19

**Rating:** 5
**Confidence:** 3

**Summary:**

This paper provides a thorough investigation of how accurate detection regions can be mapped to each other.  The detection regions are typically represented by noisy vectors with outliers, and those problems can be formulated as how to finding the matching map between two sets of d-dimensional vectors. The investigation shows that the LSL estimator is robust to noise and outliers for both problems of detection and estimation.

**Limitations And Societal Impact:**

1. It only provides theoretic analysis and experiments to study how robust of the existing approach to the noisy observation. There is no attempt to further improve the existing algorithms, such as exploring the accuracy of LSL estimator with different error measurement.
2. The author should consider the experimental evaluation with larger public known datasets.

**Main Review:**

This is an investigation which follows Collier and Dalalyan's work to explore the stability of noise observations. The paper is easy to read and understand; however, the originality and significance is more incremental.

**Time Spent Reviewing:**

2

---

> ### Author Response · Authors · 2021-08-09
> **Evaluation based on a partial overview of our contributions**
>
> We would like to thank the reviewer for the time spent reading and evaluating our manuscript. We find however that the summary included in the review provides only a partial overview of  our contributions, and the evaluation seems to be based on this partial overview. We further detail our point of view below.
> 1. According to the summary “The investigation shows that the LSL estimator is robust to noise and outliers for both problems of detection and estimation”. We would like to draw the reviewer's attention to the fact that this is not the core contribution of the paper. The core contribution is the characterization of the optimal detection region in the problem of matching. The optimal detection region is not a feature of a particular algorithm or estimator, but of the problem under consideration. It provides a precise measure of the difficulty of the problem and delineates the frontier between what is feasible and what is not. The fact that LSL is optimal is just a part of our main results. It should not be thought of as the main (or the only) take away message.
> 2. The reviewer also writes “This is an investigation which follows Collier and Dalalyan's work to explore the stability of noise observations. The paper is easy to read and understand; however, the originality and significance is more incremental.” First, we would be grateful if the reviewer could further explain what is meant by “stability of noise observations”. Second, we made an effort in presenting our results as clearly as possible and are very happy to see that this effort is appreciated. Third, we are surprised that our results are qualified as incremental. Indeed, the problem of robustness to outliers has never been tackled in the context of matching. Our main results, presented in Sections 3.2 and 3.3, provide a precise characterization of the “signal-to-noise” ratio which makes it possible to detect the matching map in presence of noise and *outliers*. The rates we obtain both in Theorem 2 and 4 are different from those obtained in Collier and Dalalyan’s work. These rates could not be inferred from prior work. Furthermore, the mathematical arguments used for finding these rates and for proving their optimality are completely different from those of [Collier and Dalalyan]. In summary, we investigated a new problem (robustness to outliers) and found new rates using new techniques. We believe that these contributions are not incremental.
> 3. Concerning the limitations of our work, the reviewer writes “It only provides theoretic analysis and experiments to study how robust of the existing approach to the noisy observation.” This is only a very partial account of our contributions; we study the case of noisy observations, *heteroscedasticity and outliers*, and show that the LSL estimator is optimal in this setting.
> 4. The reviewer also points to the necessity to “consider the experimental evaluation with larger public known datasets.” This remark is puzzling; we would very much appreciate it if the reviewer could give more details highlighting the need for experiments on a larger dataset. In experiment 3, we used a data set with 1000 pairs of images, and more than 100 vectors for each pair. This corresponds to 100000 vectors. Given that the goal of the paper is theoretical, our opinion was that there is no need to do a larger experiment.

---

### Decision · Program_Chairs · 2021-09-27

**Decision:**

Reject

**Comment:**

While this paper considers an interesting twist on the matching problem, and is well written, the reviewers seem to indicate it would be of somewhat limited practical interest due to the existence of a much more powerful method (using a much heavier model) in openCV. And on the flip side the theoretical contributions are somewhat straightforward as they are primarily the analysis of a known (non-outlier) algorithm, but when used with assumptions on the outliers.